# The structure and function of P5A-ATPases

Ping Li [1] ✉, Viktoria Bågenholm[2], Per Hägglund [2], Karin Lindkvist-Petersson [1], Kaituo Wang [2,3] & Pontus Gourdon [1,2] ✉

Endoplasmic reticulum (ER) membrane resident P5A-ATPases broadly affect protein biogenesis and quality control, and yet their molecular function remains debated. Here, we report cryo-EM structures of a P5A-ATPase, CtSpf1, covering multiple transport intermediates of the E1 → E1-ATP → E1P-ADP → E1P → E2P → E2.$P_i$ → E2 → E1 cycle. In the E2P and E2.$P_i$ states a cleft spans the entire membrane, holding a polypeptide cargo molecule. The cargo includes an ER luminal extension, pinpointed as the C-terminus in the E2.$P_i$ state, which reenters the membrane in E2P. The E1 structure harbors a cytosol-facing cavity that is blocked by an insertion we refer to as the Plug-domain. The Plug-domain is nestled to key ATPase features and is displaced in the E1P-ADP and E1P states. Collectively, our findings are compatible with a broad range of proteins as cargo, with the P5A-ATPases serving a role in membrane removal of helices, although insertion/secretion cannot be excluded, as well as with a mechanistic role of the Plug-domain.

P-type ATPases are integral membrane proteins that transport cargo across biological membranes at the expense of ATP[1]. The superfamily is divided into five subfamilies, P1–P5, based on sequence similarity and transport specificity[2,3]. P1- to P3-ATPases (such as the $Ca^{2+}$-transporting P2-member SERCA)[4] are involved in the membranous exchange of ions in all kingdoms of life, while P4-ATPases (flippases) are present in eukaryotic cells only and facilitate lipid transfer from the exoplasmic to the cytoplasmic leaflet[1]. Akin to P4-ATPases, P5-ATPases are only found in eukaryotes and are classified into two subfamilies, P5A and P5B, distinguished by preserved residues that are critical for transport[5,6]. It has been shown that P5B-ATPases carry polyamines across cellular membranes[7,8], and the overall transport mechanism was revealed by structural studies of the yeast P5B-ATPase Ypk9[9] and human ATP13A2[10–13]. In contrast to P5B-ATPases, a single P5A-ATPase is present in all eukaryotic organisms, such as ScSpf1[14,15] (in *Saccharomyces cerevisiae*), CATP-8[16] (*Caenorhabditis elegans*), AtPDR2[17] (*Arabidopsis thaliana*), and hATP13A1[18] (human), all resident in the endoplasmic reticulum (ER). Yet, P5A-ATPases have been associated with a broad palette of physiological processes. These include proper biogenesis and quality control, and in particular, correct subcellular localization of many membrane proteins to a range of different organelles, and thus also for the function of several such subcellular compartments[14,16,18–27].

All P-type ATPases share a common fold with a transmembrane (M) domain responsible for cargo transport across the membranes, and actuator (A), nucleotide binding (N), and phosphorylation (P) domains handling ATP turnover[1]. The latter involves a S/TGE dephosphorylation motif in the A-domain, as well as a conserved DKTGT stretch with the catalytic aspartate in the P-domain that becomes (de)phosphorylated. The P-type ATPase reaction mechanism, referred to as the post-Albers cycle[28,29], is classically divided into two principal states (E1 and E2) with corresponding phosphorylated forms (E1–E1P–E2P–E2, Fig. 1a). In the E1 state, the protein is in an inward open conformation. Next, binding of the transported cargo to the M-domain and nucleotide to the N-domain trigger phosphorylation of the P-domain and M-domain occlusion, E1P. The protein then converts to an outward open state, E2P, that releases the cargo and provides an opportunity for binding of 'counter' cargo, a feature that appears obsolete in certain P-type ATPase classes, while for P4- and P5B-ATPases, this cargo represents the main species transported. The P-domain is subsequently dephosphorylated by the A-domain, which is associated with re-occlusion of the M-domain leading to the E2 conformation and eventually to re-opening of the M-domain towards the interior, thereby resetting the cycle[1,30].

The P5A architecture was revealed by structures of ScSpf1 intermediates covering two distinct states: an inward (cytosol)-open and an

[1]Department of Experimental Medical Science, Lund University, Sölvegatan 19, SE-221 84 Lund, Sweden. [2]Department of Biomedical Sciences, University of Copenhagen, Nørre Allé 14, DK-2200 Copenhagen N, Denmark. [3]Present address: State Key Laboratory of Plant Diversity and Specialty Crops, Institute of Botany, Chinese Academy of Sciences, Beijing, China. ✉e-mail: ping.li@med.lu.se; pontus.gourdon@med.lu.se

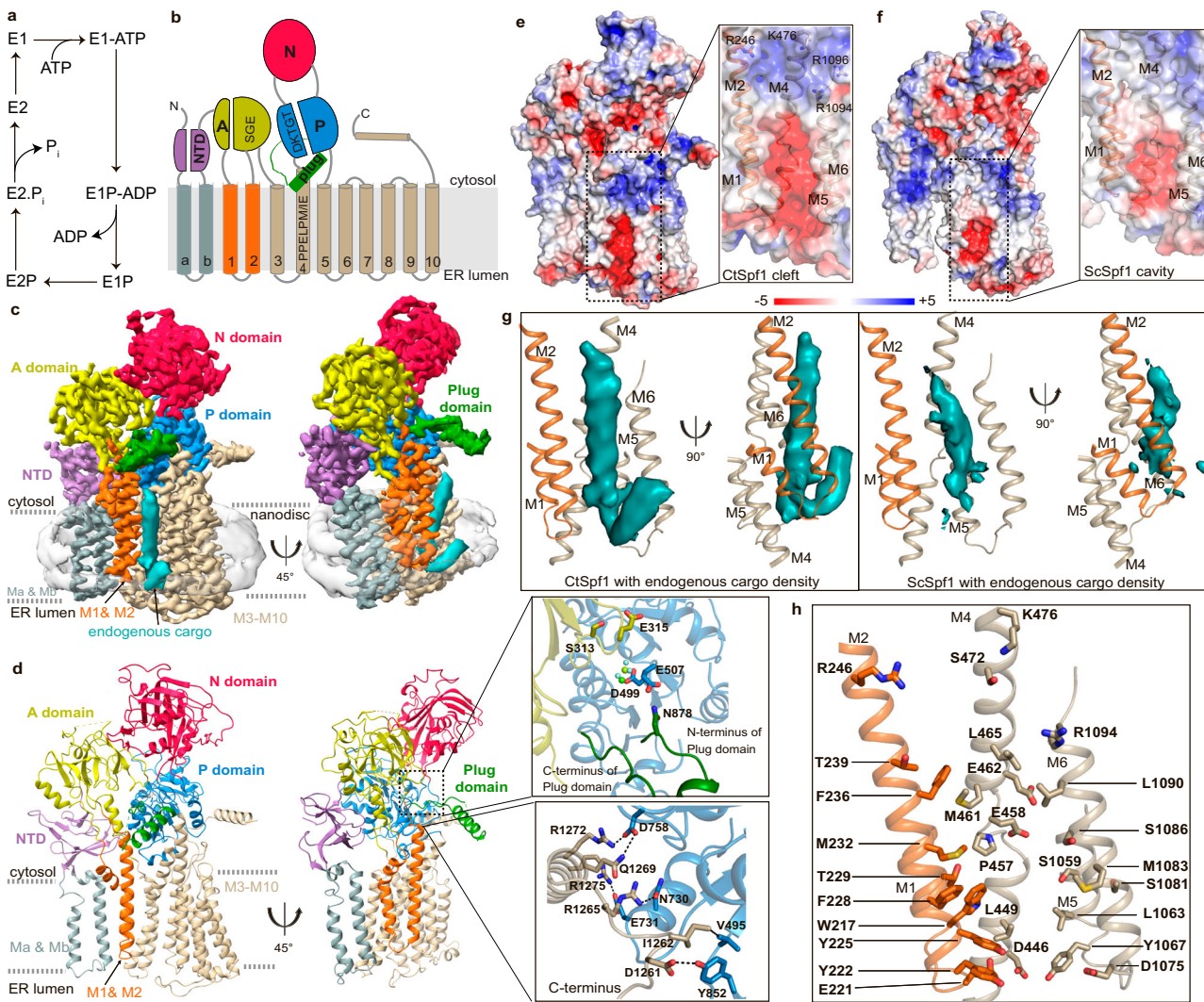

**Fig. 1 | Overall architecture of the P5A-ATPase CtSpf1 in the E2P state with bound endogenous cargo.** Note the cargo helix is not modeled in E2P, and hence the cleft appears more exposed than it is in the structure. **a** The P-type ATPase transport cycle follows the E1–E1P–E2P–E2 post-Albers scheme. **b** P5A-ATPase topology with the A- (colored yellow), P- (blue), N- (red), M- (helices Ma/Mb in gray, M1/M2 in orange and M3-M10 in wheat), NTD- (purple) and Plug- (green) domains. **c** 3.5 Å global resolution cryo-EM map, colored as in panel (**b**), with a co-purified polypeptide (cyan, non-sharpened) and nanodisc (transparent gray). **d** Cartoon representation, colored as in panel a, and in the same view as panel (**c**). Close-view of the interaction of the P-domain insertion with the P-domain and SGE dephosphorylation of the A-domain (upper panel); and close-view of the interaction of the C-terminus with the P-domain (lower panel). **e** Surface electrostatics of CtSpf1, with a close view of the cleft spanning the entire ER membrane formed primarily by M1, M2, M4, and M6. **f** Equivalent surface electrostatics for ScSpf1 (PDB-ID 6XMT[23]), with a cavity facing the ER lumen. **g** Uncropped cryo-EM density (cyan, left panel) of the cargo helix associated with the M-domain cleft CtSpf1, and the equivalent for the cavity present in ScSpf1 (cyan, left panel, EMD: 22264). **h** Manually selected residues that may interact with the cargo. Residues highlighted in bold are conserved among P5A-ATPases.

outward (ER)-open conformation[23] (Supplementary Table 1). In addition to the classic P-type ATPase fold, these structures established features unique to P5A-ATPases, including an N-terminal domain (NTD), composed of a soluble β-sheet that interacts tightly with the A-domain, and two class-specific transmembrane helices[23] (Ma and Mb) (Fig. 1b). Ma and Mb are exclusive to P5A-ATPases (no similarity to the pre-core transmembrane segments found in P1B-ATPases[31]), and the NTD is different also to that in P5B-ATPases[9,32,33]. Another unique feature was an insertion in the P-domain in proximity to M5, protruding from the core (a considerably shorter extension is present in P5B-ATPases, Supplementary Fig. 1)[23]. In the ER-open structures, an endogenous polypeptide was caught in the transport pathway of the M-domain. Yet, it remains elusive how the transport of such a large cargo is achieved and how specificity is established. It has been proposed that P5A-ATPases serve in quality control of proteins, removing (dislocating) certain mistargeted (intended for another compartment)

or mis-inserted (with inverted topology) proteins from the ER membrane[15,23,34–39], with the cargo representing the 'counter' cargo in classical P-type ATPases, alike the situation in P4- and P5B-ATPases. A similar function, but with consequences on a large portion of the membrane protein proteome, has also been suggested[40]. However, there are also studies indicating an alternative yet interlinked, equally broad role for the P5A-ATPases in membrane insertion of proteins[16,18,19,22,41,42]. Moreover, ScSpf1 has been linked to sterol, Ca²⁺, Mn²⁺, or lipid homeostasis[14,43–46], protein degradation[34], and represents one of few proteins that influence the orientation of type II membrane proteins (with a single transmembrane helix and the N-terminus normally located in the cytosol)[35,47]. All in all, this leaves fundamental questions regarding the P5A in (patho)physiology, from the cellular to the molecular level.

Here, we provide a series of single-particle cryo-electron microscopy (cryo-EM) structures of a P5A-ATPase with and without captured

endogenous cargo, complemented by mass spectrometry. Our combined data shed light on the complexity of this elusive class of P-type ATPases.

# Results

## Overall structure

To further illuminate the P5A transport mechanism, we selected the P5A-ATPase homolog CtSpf1 from the thermophilic fungi *Chaetomium thermophilum*[48], which shares 54% sequence identity with ScSpf1 (Supplementary Fig. 1). Throughout, the target was studied using C-terminally fused green fluorescent protein (GFP), overproduced in *S. cerevisiae*. We performed a *S. cerevisiae*-based in vivo assay to assess the function of CtSpf1. Akin to ScSpf1, CtSpf1-GFP is targeted to the ER membrane (Supplementary Fig. 2a). Moreover, expression of wild-type CtSpf1-GFP in a *Spf1* knock-out *S. cerevisiae* strain rescues the cells from caffeine sensitivity, which is not the case for a catalytically inactive mutant of CtSpf1-GFP (Supplementary Fig. 2b). These observations are consistent with previous studies of ScSpf1[8,43,49] and show CtSpf1 can complement ScSpf1 in vivo.

For structure determination, CtSpf1-GFP was purified using a mild isolation strategy to potentially capture endogenous cargo, which included the addition of state-catching agents already prior to cell lysis, and the sample, including GFP-tag was eventually reconstituted into nanodiscs for single particle cryo-EM studies (Methods). We first stabilized the sample using the phosphate mimic $BeF_3^-$. Based on this preparation, a cryo-EM structure was determined at an overall resolution of 3.5 Å, yielding a cryo-EM map with well-resolved domains, except for certain peripheral loops (Fig. 1c, d, Supplementary Fig. 3 and Supplementary Table 2). The structure harbors the classical P-type ATPase architecture, with cytosolic A-, P-, and N-domains and a transmembrane M-domain with a core of ten helices (M1–M10). As seen previously[23], the fold also contains the NTD, as well as the insertion in the P-domain that is unique to P5-ATPases (Supplementary Figs. 1 and 4). The latter stretches from residue L877 to G1013, structurally intercalated in between the A- and P-domains. In the C-terminus, M10 is followed by a short loop and a rigid helix, which interacts with the P-domain akin to certain P5B-ATPases such as CtYpk9[9]. It is possible that this arrangement permits regulation of turnover, a frequently observed feature for flanking stretches of P-type ATPases[9,50–52].

## An E2P state with a cleft that spans the entire membrane

$BeF_3^-$ is known to trap E2P states of P-type ATPases characterized by an "outward-open" M-domain, to ensure cargo release (and/or counter cargo uptake), and a tight configuration of the soluble domains to enable dephosphorylation[1,53,54]. Accordingly, in our structure, the conserved SGE dephosphorylation loop (ending with E315) of the A-domain is located about 6 Å from the invariant catalytic aspartate of the P-domain (D499), and we identify the phosphate mimic in the cryo-EM density essentially bridging these key features (Supplementary Figs. 4 and 5a). The N- and C-termini of the P-domain insertion, which forms an extended helix and a loop, respectively, are adjacent to each other as they emerge from the ATPase core, both projecting into the surrounding environment (Fig. 1c, d, Supplementary Figs. 3 and 6). While only the P-domain insertion connections are possible to model, most or all of the region is visible in non-sharpened cryo-EM maps of the structure, indicating that the initial helix is followed by a globular domain that dips toward the membrane interface.

In the M-domain, we observe a funnel-shaped cleft spanning the entire membrane, which, if no cargo was bound (see below), would be directly exposed to the surrounding hydrophobic environment (Fig. 1e). It is lined primarily by helices M2, M4, and M6, with contributions also from M1, M5, and M8, exposing numerous highly conserved residues of M1, M2, and M4. The fissure displays a considerable opening towards the ER lumen, which is partially capped by the loop

connecting M7 and M8. That the cleft spans the entire membrane differs from the ScSpf1 E2P state (RMSD 2.8 Å to PDB-ID 6XMT[23], Supplementary Table 1), which adopts an ER lumen facing cavity, overlapping with the most voluminous portion of the CtSpf1 cleft (Fig. 1f, g). This discrepancy relates to the arrangement of the invariant P5A motif PPELPM/IE of M4, which holds a more peripheral location in ScSpf1 (Supplementary Fig. 4a). Both the ScSpf1 cavity and the CtSpf1 cleft are electronegatively charged, as facilitated by the P5A-invariant E458 and E462 of PPELPM/IE, located towards the cytosolic interface, but also hosts uncharged amino acids (Fig. 1e, f). In contrast, the CtSpf1 aperture at the interface towards the cytoplasm is somewhat electropositive due to R246 (of M2), K476 (M4) and R1094/R1096 (M6). The opening towards the ER lumen is instead strongly electronegative because of E221 in the M1/M2 loop, D446 (M4), D1075 (M6), as well as E1150 and E1152 (M7/M8). With the exception of R246 and E1150, these mentioned charged amino acids on opposite sides of the membrane are relatively well-conserved in P5A-ATPases.

## Bound cargo

Perhaps the most striking result from the previous ScSpf1 structures was the detection of a membrane-spanning helix in the E2P state, largely located in the same M2, M4, and M6 region as the polyamines transported by P5B-ATPases[9,23]. In our structure, non-ATPase cryo-EM density passing through the entire membrane is observed that overlays with the cargo identified in ScSpf1, but it stretches further in both directions (Fig. 1c, g). The feature reaches the ER lumen and then bends back into the hydrophobic phase, peripheral to the ATPase. On the other side of the membrane, it is exposed to the cytosol and almost reaches the P-domain. As permitted by the displacement of PPELPM/IE, the density is more deeply buried into the cleft and displays a subtle difference in its angle relative to the M-domain, compared to ScSpf1. Owing to a lack of clear secondary structure, we prefer not to model the feature. However, our interpretation is that it represents an average of endogenously transported cargo co-purified with CtSpf1, and hence we denote the conformation E2P$^{cargo}$. Numerous residues in M1 to M5, as well as in the M7/M8 loop, directly point to the cargo and are likely involved in binding (Fig. 1h). A moderately hydrophobic cargo helix and an associated, positively charged ER lumen flanking C-terminal sequence has previously been indicated to be critical for P5A-interaction with so-called tail-anchored (TA) membrane proteins destined for mitochondria[23,38,55]. However, the cargo density we observe is well-integrated in the cleft, and the E458 and E462 key residues interact with the helix part instead of a tail. In addition, the polypeptide loop reenters the membrane environment, indicating this part is uncharged. Therefore, it is likely the cargo we captured in the E2P state represents other client types or only a small fraction of mis-inserted membrane proteins with positively charged C-termini, such as mitochondrial TA proteins. Indeed, type II membrane proteins that also have been proposed as clients for P5A-ATPases better agree with the observed cryo-EM density[38].

## The transition state of dephosphorylation (E2.Pi) with endogenous cargo

An alternative phosphate mimic, $AlF_4^-$, stabilizes P-type ATPases in a consecutive "outward-facing" transition state of dephosphorylation (E2.P$_i$), a conformation that has never been structurally characterized for a P5A-ATPase (Supplementary Table 1). Treating CtSpf1 with $AlF_4^-$ (Methods) yielded one structure, determined at 3.4 Å overall resolution, with the phosphate mimic present and a high similarity to the above described E2P state (RMSD 0.5 Å, Fig. 2a, b, Supplementary Figs. 5b and 7, Supplementary Table 1). Nonetheless, the N-domain is rearranged compared to the E2P state, although to a smaller extent than in P2-ATPases and E315 is positioned somewhat closer to the phosphate (mimic) bound to D499, likely preparing the protein for dephosphorylation. The location of the P-domain insertion appears

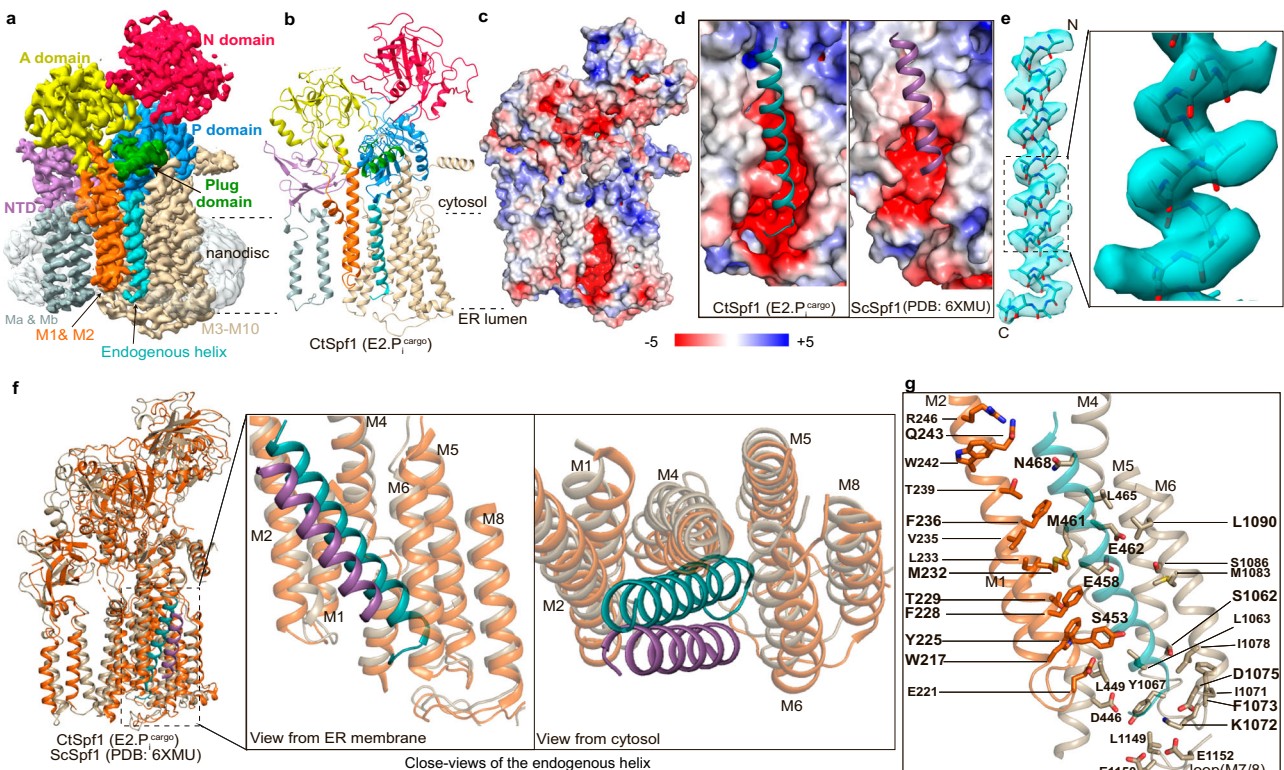

**Fig. 2 | The E2.Pi state structure with bound endogenous cargo. a** 3.4 Å global resolution cryo-EM map, colored as in Fig. 1b. **b** Cartoon representation, with the cargo helix modeled as a poly-alanine (cyan). **c** Overview of the surface electrostatics without the modeled helix and with a similar view as Fig. 1e. **d** Close-views of the cleft and cavity, respectively, including the poly-alanine chain bound to CtSpf1 (cyan) and ScSpf1 (purple). **e** Cryo-EM density of the captured cargo helix, built as 31 alanines, presumably with the N-terminus in the cytoplasm (note the Christmas tree effect). The close view shows side-chain densities. **f** Structural comparison of cargo bound to CtSpf1 (E2.Pi^cargo, wheat, cargo in cyan) and ScSpf1 (PDB-ID: 6XMU, orange, cargo in purple) and close-views of the corresponding M-domains. **g** Residues potentially involved in binding to the cargo helix (all within 5 Å from the modeled polyalanine are shown). Residues highlighted in bold are conserved among P5A-ATPases.

basically unaltered compared to E2P^cargo (Supplementary Fig. 6). The membrane spanning cleft is essentially maintained too, and cargo remains present, leaving us to designate this configuration E2.Pi^cargo (Fig. 2c, d). However, the cargo density demonstrates clearer helix features than in E2P^cargo, but lacks the loop-style bent tail. While the exact sequence of the cargo still cannot be assigned, clear density is available for certain sidechains, positioning 31 residues (as poly-alanine) in the final model (Fig. 2e). The helix cargo remains buried in the membrane-spanning cleft (Fig. 2b, e, f), like in E2P^cargo. Notably, we modeled the N-terminus in the cytoplasm because the sidechains generally point in that direction. Contrary to the E2P state, this arrangement could be in agreement with the presence of e.g., mis-inserted mitochondrial TA proteins or even mis-inserted multi-spanning membrane proteins such as the equivalent of ABCG2 in humans[38,39]. As the cargo is more defined in E2.Pi^cargo the details of the interaction network with the cargo can be further scrutinized. Multiple conserved residues from critical P5A-motifs, including W217 (of M1), Y225, F228, M232 and Q243 (M2), S453, E458, E462, and N468 (M4), S1062, K1072, and F1073 (M5/M6) as well as D1075 and L1090 (M6) are within 5 Å of the modeled helix (Fig. 2g).

## E1 configurations with an inaccessible inward-facing cavity

Subsequently, we also recovered an apo CtSpf1 structure, isolated without the presence of state-stabilizing agents or inhibitors (Methods), determined at an overall resolution of 3.4 Å with well-resolved domains (Fig. 3a, Supplementary Figs. 8 and 9). Structure-based alignments of the structure to P2-, P4-, and P5-ATPases indicate it resembles an E1 or E1-ATP state, which is typically associated with

"inward-open" or "inward-facing occluded" configurations, respectively (RMSD 5.5, 5.0, 5.7, 2.2, and 3.0 Å to PDB-IDs 4H1W[56], 3N8G[57], 6K7G[58], 6XMP[23], and 7N75[13], respectively, Supplementary Fig. 10 and Supplementary Table 1). While the relative positions of the N- and P-domains in these states are preserved across these P-type ATPase classes, the observed location of the A-domain varies and is here distinct from that of P4- and even more so P2-ATPases, being instead rather reminiscent of P5B-ATPases. This places the SGE-motif in-between the two ends of the P-domain insertion (residues 877–878 and 1013–1017), with the N-terminal end also being adjacent to the DKTGT-motif, thereby nestling key aspects of the core of P-type ATPases with this P5A-specific extension (Fig. 1d).

In the M-domain, we find a deep cytosol-facing crevice, in-between primarily M2, M4, and M6 but partially lined also by M1 and M8, that deviates substantially from the cleft identified in the E2P/E2.Pi states. The cavity, together with the arrangement of the soluble domains, renders us denoting the state E1. The identified inward-open crevice likely represents a critical aspect of the transport pathway, since the opening is overlaying the cleft of the M-domain in the E2P/E2.Pi states, and the cleft is lined by E458 and E462 (Supplementary Fig. 5d). However, the cavity is blocked by the C-terminal portion of the P-domain insertion, with residues 1005–1013 dipping into the membrane and inserting between M2 and M4, hence forming another link between the soluble and M-domains (Fig. 3b, c and Supplementary Fig. 6). Concurrently, the main part of the P-domain insertion has rearranged somewhat compared to E2P/E2.Pi (Supplementary Figs. 5d and 6). Moreover, the apo cryo-EM data yielded two maps of the same resolution, where the only difference is the relative

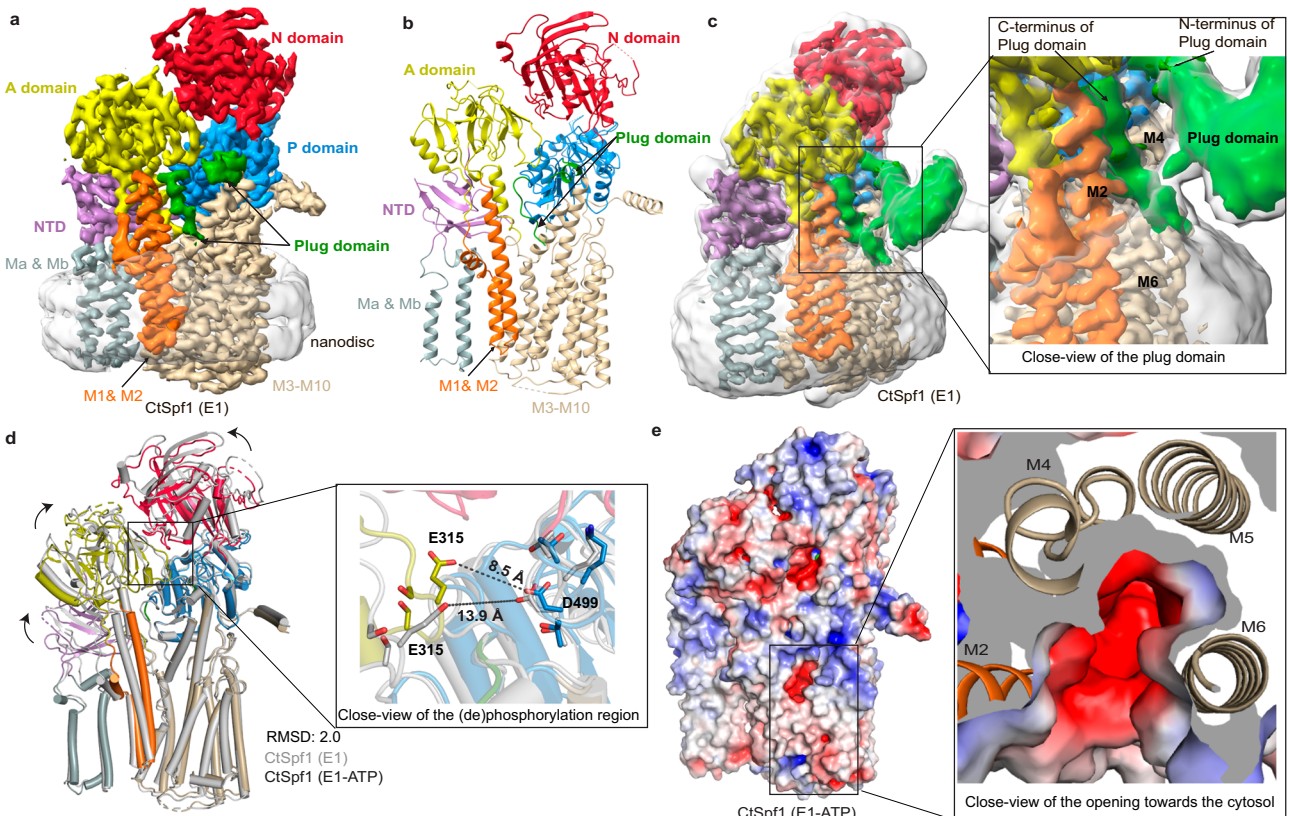

**Fig. 3 | The blocked cytosol-facing apo E1 and E1-ATP states. a** 3.4 Å global resolution cryo-EM map of the E1 structure, colored as in Fig. 1b. **b** Cartoon representation of the E1 state. **c** The Plug-domain (green) dips into and blocks the cytosol-facing cavity, inserted in between M2 and M4. Non-sharpened, Cryo-EM map (gray, transparent) at contour level 0.13, and non-sharpened density for the Plug-domain (green) at contour level 0.16. **d** Structural comparison of the apo E1 (gray) and E1-ATP states (colored domains) highlighting the movement of the SGE dephosphorylation loop. **e** Surface electrostatics of the E1-ATP state, with a deep cavity in the M-domain that is exposed to cytosol, unless blocked by the Plug-domain, which is not modeled. The close view is from the cytosol, showing that the cavity is lined by M2, M4, M5, and M6.

position of the P-domain insertion relative to the ATPase core, revealing additional flexibility (Supplementary Fig. 6). The observed ability of this P-domain insertion to block the likely transport pathway in the M-domain lead us to rename it the Plug-domain and the C-terminal residues of the domain (residues 1005–1013 model, but non-sharpened maps suggest it extends beyond D1005) the Plug, the significance of which will be discussed later.

Non-hydrolysable nucleotide derivatives have previously been exploited for isolation of "inward-facing" transition states of phosphorylation of P-type ATPases[4,9,23,59]. Using one such compound, AMP-PNP, we recovered a structure determined at an overall resolution of 3.2 Å, but most core regions reached 2.8 Å, with cryo-EM density supporting binding of the state-stabilizer (Supplementary Figs. 5a and 11). The overall configuration is similar to the E1 structure (RMSD 2.0 Å, Supplementary Table 1), including the cavity facing to the cytosol (Fig. 3d, e). However, a somewhat more compact arrangement of the soluble domains, again like that observed in P5B-ATPases of the equivalent state, leaves us denoting the configuration E1-ATP. This places the nucleotide and the associated $Mg^{2+}$ close to D499, preparing the ATPase for phosphorylation. At the same time, E315 is relatively close to D499, which appears to be a P5-specific feature linked to the Plug domain. The Plug-domain is unaffected overall compared to the E1 configuration. Yet the C-terminal portion appears more flexible (only residues 1012–1013 were modeled), and hence cytoplasmic access cannot be excluded in E1-ATP (Supplementary Fig. 6). In the M-domain, M1 and, in particular, M2 are approaching M4 and M6, thereby narrowing the cleft somewhat, but other parts of the M-domain remain essentially unaltered (Fig. 3d).

## Inward-open states with features adjacent to and in the inward-facing cavity

To capture a subsequent E1 transition state of phosphorylation, E1P-ADP, we incubated the CtSpf1 sample with $AlF_4^-$ throughout cell lysis and purification, and then also with ADP prior to grid preparation, a combination that traps an inward-facing transition state of phosphorylation of P-type ATPases[60]. We retrieved two separate maps with ADP-$AlF_4^-$ bound, with overall identical conformation that are both reminiscent of our E1-ATP structure (Fig. 4a, Supplementary Figs. 5e and 12, Supplementary Table 1). Indeed, the overall arrangement of the domains, the nucleotide-binding region, and the cavity exposed to the cytoplasm are maintained, and we denote the conformation E1P-ADP. However, as deduced also from non-sharpened maps, the main part of the Plug-domain has adopted an entirely new location compared to E2P/E2.$P_i$ and E1/E1-ATP. The domain has swung away from the membrane interface and with no remaining indications of the cleft being blocked, despite overall small changes of the soluble domains relative to E1 and E1-ATP (Supplementary Fig. 6). Interestingly, the major difference between the two maps is that one of the cryo-EM maps harbors an elongated cylinder-shaped density that spans most of the membrane (Fig. 4a). The feature is located close to the membrane interface, where the cytoplasmic end of the density is adjacent to the opening of the M-domain cavity, and with the rest of the density tilted away from the protein, all peripherally yet in close proximity to the above-mentioned cavity (Fig. 4b, c). The closest residues on the edge of the cavity are hydrophobic (Fig. 4d). This deviates from the E2P/E2.$P_i$ structures with bound cargo, in which the cargo is deeply buried in a cleft. We refrain from modeling the feature

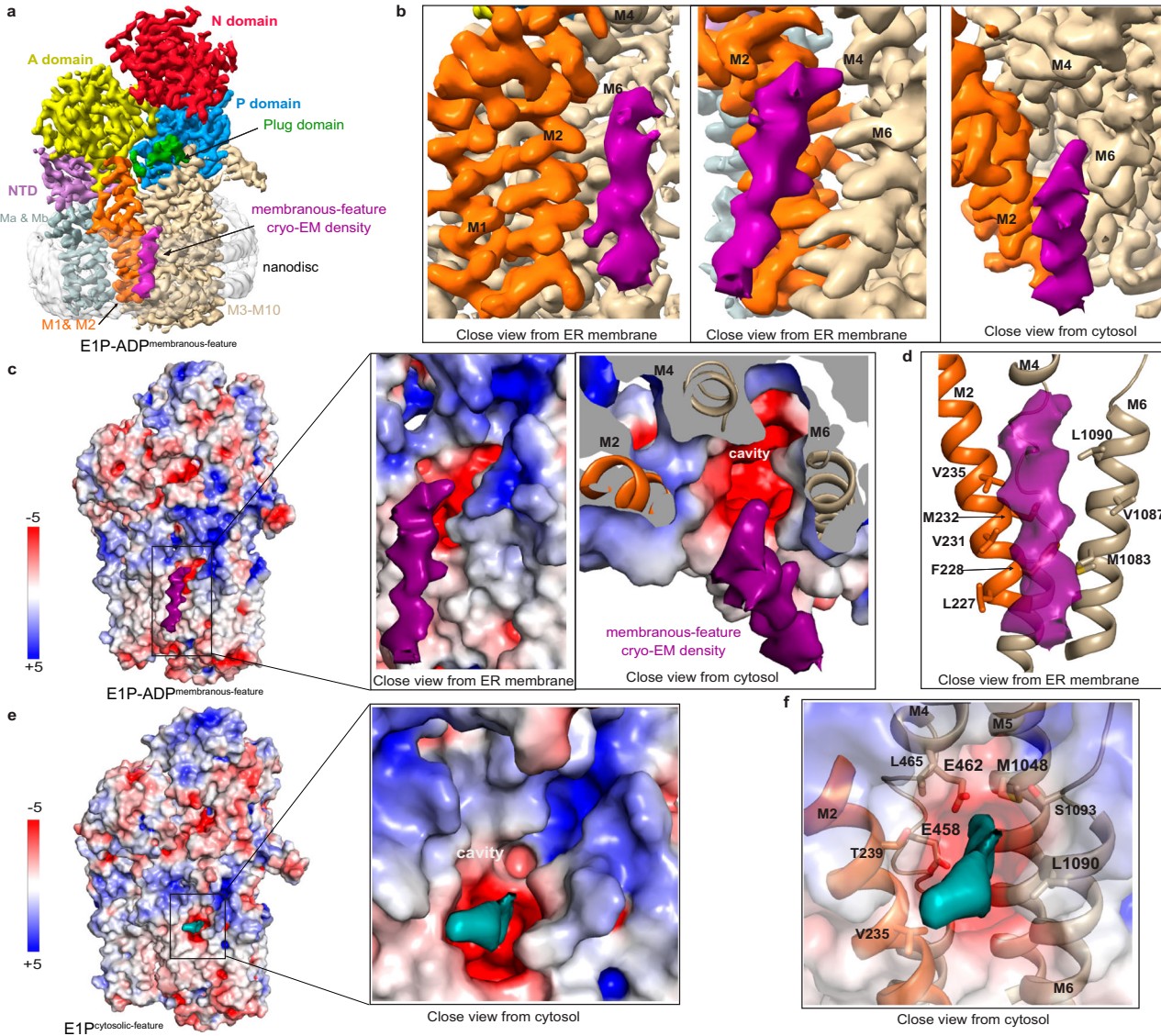

**Fig. 4 | The cytosol-open E1P-ADP and E1P states with associated cryo-EM density features. a** 3.4 Å global resolution cryo-EM map of the E1P-ADP state, colored as in Fig. 1b, with an associated unknown, elongated cylinder-shape density (pink), located peripherally to the cargo helix observed in E2P/E2.P$_i$. **b** Close-views of the unknown E1P-ADP density (pink), spanning the entire membrane and associating with M2 and M6. **c** Surface electrostatics of the E1P-ADP structure, including close views of the unknown density (pink) associated with the hydrophobic surface.

**d** Manually selected residues (sticks) that may interact with the unknown feature (pink). **e** Surface electrostatics of the E1P state, with a deep pocket in the M-domain that is exposed to the cytosol and a strong cryo-EM density that is located at the intracellular membrane interface of the cavity (cyan). **f** The E1P density at the cavity opening (cyan). Manually selected residues (sticks) that may be involved in binding to the feature.

due to the poorly defined secondary structure and refer to the structure as E1P-ADP$^{membranous-feature}$. Although we expect the observation to relate to a helix due to its size and shape, it cannot be excluded that a lipid has been caught.

Finally, we deduced another state from the cryo-EM dataset employed to determine the E2.P$_i$ structure, stabilized using bound AlF$_4^-$ only, where the nucleotide binding pocket is unoccupied (Supplementary Figs. 5a and 7). It is structurally homologous to the above-mentioned E1P-ADP structure, maintaining also the overall configuration of the Plug-domain, and the corresponding form of P4- and P5B-ATPases has been assigned as E1P[13,58] (Supplementary Figs. 5e and 6, Supplementary Table 1). This is coupled to a strong non-ATPase density positioned at the intracellular membrane interface, which is stretching into the cytosol-facing cavity that may relate to parts of the cargo, and therefore we refer to this state as E1P$^{cytosolic-feature}$ (Fig. 4e). Multiple amino acids directly point to the density, including the

conserved E458 and E462 (of M4), M1048 (M5) and L1090 (M6), indicating these residues may be involved in binding of the feature (Fig. 4f). The unmodelled density in this structure is located adjacent to the unknown density of E1P-ADP$^{membranous-feature}$, and it is also less extended. Thus, the feature may correspond to cargo in transit, but we cannot rule out that it rather represents some smaller molecule. Due to the large similarity to E1P-ADP$^{membranous-feature}$ it can also not conclusively be determined which of these structures would precede the other in the cycle.

## ATP turnover is coupled to M-domain reconfiguration and Plug-domain gating

How can then P5A-ATPases provide transport of such large cargo? The cargo-filled all-through membrane spanning cleft and the cytosol-facing cavity, accompanied by the shifts of the Plug-domain, are key for understanding this. Unbiased analyses of the relative locations of

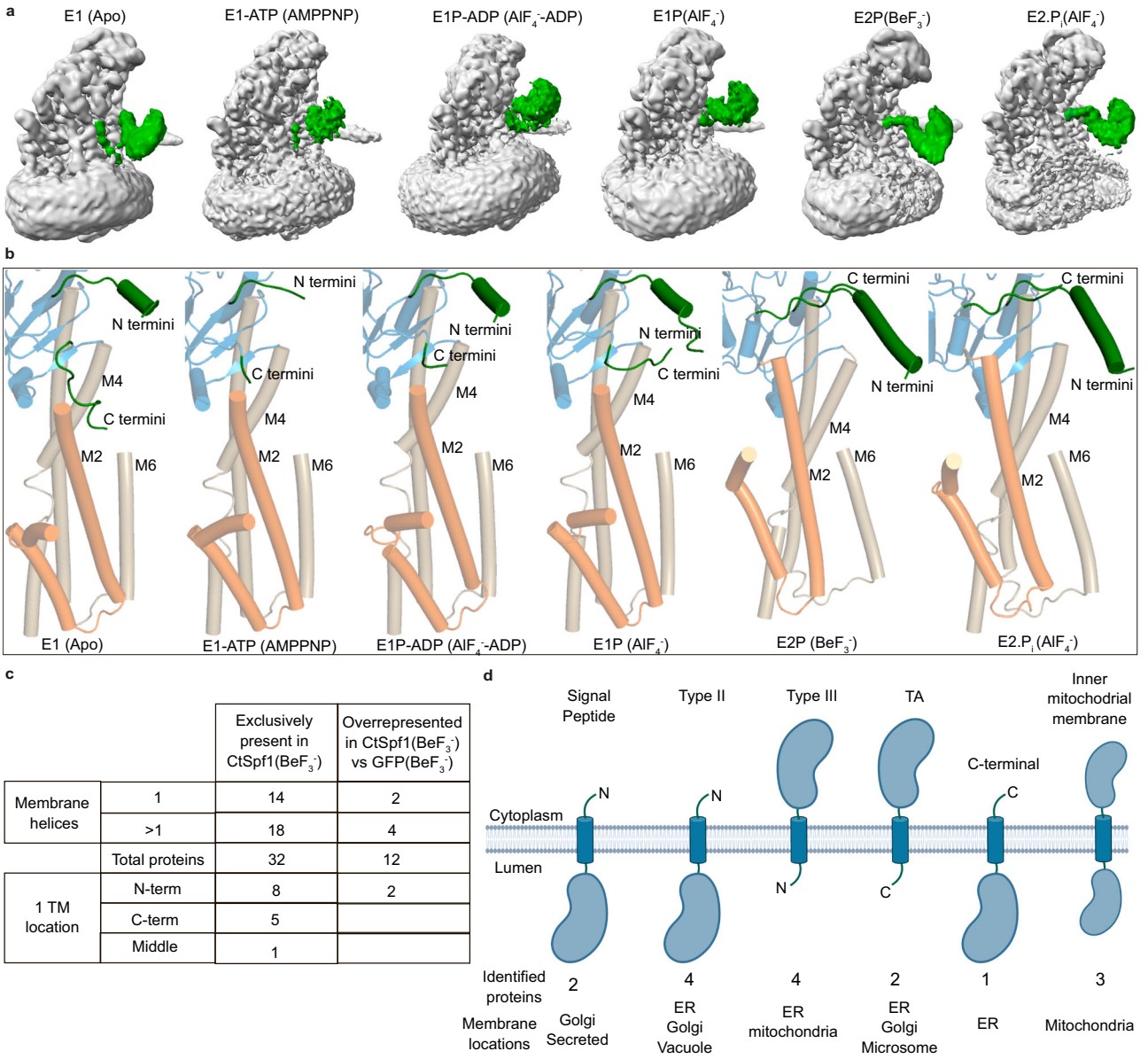

**Fig. 5 | The conformational changes of the Plug-domain in different P5A states and mass-spectrometry-identified possible cargo. a** Non-sharpened cryo-EM map of the here determined states. The Plug-domain is highlighted in green and the CtSpf1 and the nanodiscs are in gray. **b** Close-views displaying the conformational changes of the modeled parts of the Plug-domain (green), the connection to the ATPase core (blue), and the TM domain (orange and wheat). **c** Overview table of proteins with at least one TM helix (including signal peptides, SP) identified with MS. Hits exclusively found in CtSpf1 in the presence of BeF$_3^-$ and not in controls, and hits among the 20 most overrepresented in the same CtSpf1 (BeF$_3^-$) sample compared to GFP in the presence of BeF$_3^-$ have been included. Two SPs were identified, both exclusively found in CtSpf1(BeF$_3^-$). The table distinguishes between identified proteins with one or multiple TM helices (for a detailed list of these hits, see Supplementary Table 3), and specifies the in-protein location of the TM helix in the identified 1 TM proteins. It is likely that some of these hits represent transport cargo for CtSpf1. **d** Overview of the identified proteins containing an SP or one TM helix, displaying the overall topology (when correctly inserted), the number of identified hits, and the membrane into which the hits are inserted. TA denotes non-mitochondrial TA proteins. For a complete list of the proteins, see Supplementary Table 3. Figure 5d was generated using BioRender; https://BioRender.com/n26e116 - Gourdon, P. (2024).

the transmembrane helices in the separate states reveal displacements primarily occur in blocks of helices, Ma–M2 relative to M4–M10, separated by M3, supporting that M1 and M2 orchestrate the overall configuration of the M-domain, and that Ma and Mb are extensions that come along M1 and M2 (Supplementary Fig. 13). The relatively small shifts of the transmembrane helices can be explained by the equivalently modest changes of the soluble domains compared to P2-ATPases. This further emphasizes the role of the A-domain for turnover, as it is directly linked to M1 and M2, and is more intimately linked to the P-domain throughout the transport cycle as observed in our structures.

The function of the Plug-domain has remained elusive, and yet it is omnipresent among P5A-ATPases, but not available in other P-type classes, implying P5A-specific significance. Indeed, it has been shown that the Plug-domain is critical for autophosphorylation of ScSpf1, but not essential for ATP hydrolysis[61]. We show the Plug-domain blocks the transport pathway in the E1 (the E1-ATP state may already be open), but not in the E1P-ADP/E1P and E2P/E2.P$_i$ configurations (Fig. 5), hence providing an additional, transport mechanism-dependent, gate-keeper level. Overall, the Plug-domain is poorly conserved in P5A-ATPases, except for the immediate linkers to the P-domain detected in our structures, suggesting these connections are important for function.

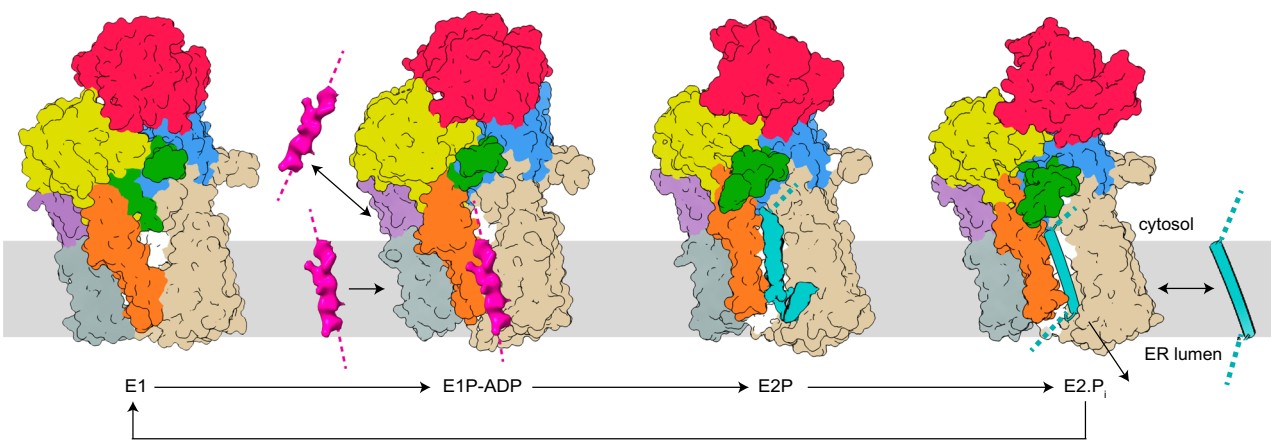

**Fig. 6 | Proposed polypeptide translocation mechanisms of P5A-ATPases.** The domains are colored coded as in Fig. 1b. P5A-ATPases seemingly rest in a cytosol-facing E1 state, which is blocked by the Plug domain (green). Autophosphorylation leads to displacement of the Plug-domain, and hence the cytosol-facing cavity becomes accessible to the cytosol in the E1P–ADP/E1P conformations. The unknown features (pink) in the latter structures may represent cargo (1) already removed from the ER membrane (in E1P, not shown in the model), and about to be bound for a new dislocation-cycle, or (2) signal-peptide, preparing the following peptide-stretch for membrane insertion. Polypeptide binding (cyan) in the all-through cleft is achieved in the E2P and E2.$P_i$ states. The clearer secondary structure in E2.$P_i$ may be (1) coincidental, or (2) an indication of folding of the inserted helix. Completion of the cycle, dephosphorylation, and reestablishment of the E1 state is associated with the release of the helix to (1) the cytosol as a removed helix or (2) the surrounding membrane or ER lumen as an inserted or secreted helix.

This is further corroborated by the C-terminal end of the Plug-domain, L1010–G1013 (which is visible in all cryo-EM maps), displaying large conformational changes between different states, obstructing the cytosol-facing cavity in the E1 and perhaps also the E1-ATP state. Indeed, the N-terminus of the Plug-domain forms an extended α-helix in experimental and AlphaFold models of P5A-ATPases, in agreement with this secondary structure element serving as a hinge that controls the location of the Plug(-domain). The fact that Plug-domain shifts were not observed in the ScSpf1 E1/E1P structures is perhaps due to the equivalent features being tightly linked to the DDM micelle through-out, preventing the shifts of the Plug-domain detected in CtSpf1[23]. It also cannot be excluded that different temperature optima of the two Spf1 variants are responsible for the lacking structural differences of e.g., the Plug-domain in ScSpf1, although it would perhaps rather be expected that smaller changes would be detected for CtSpf1 than for ScSpf1 at the employed experimental conditions.

Moving through the cycle, we propose ATP binding and initiation of autophosphorylation induces the N-domain to approach the P-domain, accompanied by modest shifts of the A- and NTD-domains that control the aperture of the cytosol-facing cavity, via displacements of M1 and M2 (from E1 to E1-ATP, Fig. 6). Associated, the Plug and Plug-domain start to rotate away from the cavity opening, as perhaps facilitated by the conserved G1013 and the A-domain rotation. Next, features that may relate to cargo are observed next to or in the pocket, in the membrane (E1P-ADP^membranous-feature), or intracellularly (E1P^cytosolic-feature). Once fully phosphorylated and converted to the E2P state, the N-domain is instead removed from the P-domain. Moreover, the A-domain is shifted to place the SGE motif nearer the D499, which remodels the transport pathway from a cytosol-facing cavity to a transmembranous cleft, via shifts of M1 and M2. The entire Plug domain is displaced relative to the ATPase core and approaches the membrane interface, which resonates with the nanodiscs becoming distorted around the Plug, creating direct access to the cargo-containing membrane penetrating cleft. This places the CtSpf1 Plug-domain in approximately the same location as observed in the ScSpf1 structures. Finally, the protein reverts to the E2 state, accompanied by the completion of dephosphorylation, and reaches the E1 conformation. The arrangement of the PPELPM/IE-motif of M4 regulates cargo binding; its location towards the ATPase core (as in our E2P and E2.$P_i$ states) permits the buried location of the cargo, while the peripheral placement may ensure a looser interaction (as in the BeF$_3^-$ stabilized ScSpf1 structure) and complete loss of cargo (as in our E1 structure). Congruent with previous reports, a membranous cryo-EM feature present between Ma, M1, and M2 in all our E1 but not the E2 structures may support that P5A-activity is regulated by lipids[43,62] (Supplementary Fig. 14).

Collectively, our data point towards an intimate and unusual arrangement of the A-, P-, and Plug-domains, underscoring the uniqueness of P5A-ATPases and their transport mechanism compared to other classes of the superfamily. The termini of the Plug-domain are sandwiched between essential core portions of the A- and P-domains, which enable conformational changes of the Plug-domain, M1, M2, and M4, that are coupled to ATP turnover and the classical P-type ATPase cycle. Thus, the plug domain orchestrates access to the transport pathway and may even be involved in the pushing/pulling of the cargo.

## On the cargo type of P5A-ATPases

It has recently been suggested that P5A-ATPases serve as ER membrane transmembrane helix dislocases, removing mis-targeted mitochondrial TA-proteins and/or related topology control of other membrane proteins, such as mis-inserted type II membrane proteins[23,37–39]. Here we demonstrate cargo captured in the E2P and E2.$P_i$ states, and likely also in the E1P-ADP^membranous-feature configuration. In the E2P and E2.$P_i$ conformations, the cargo is relatively long and deeply buried in the cleft, with helix segments placed immediately adjacent to the negatively charged E458 and E462. Clear helix density supports that the N-terminus is located in the cytoplasm in the E2.$P_i$ structure. In addition, an ER-located tail reentering the membrane was detected in the E2P structure, which is relevant as many proposed P5A-ATPase clients have positively charged tails in the ER lumen when mis-inserted. Collectively, it thus appears likely that broad types of membrane proteins would be transported by P5A-ATPases.

To attempt to shed further light on the transported cargo of P5A-ATPases, mass spectrometry was carried out analyzing purified CtSpf1 with or without the phosphate mimic BeF$_3^-$, reminiscent of the conditions used for structure determination of the E2P^cargo and apo E1 states, respectively. Constructs only overproducing soluble GFP were also run as controls. Thus, proteins that are either identified exclusively in CtSpf1/BeF$_3^-$, or overrepresented in CtSpf1/BeF$_3^-$ relative to the GFP and CtSpf1-apo controls, constitute hypothetical clients of

P5A-ATPases (Fig. 5c & Supplementary Table 3). However, we did not find a single mitochondrial TA-protein among the potential clients. In general, for identified proteins containing at least one transmembrane segment or signal peptide, data from UniProt[63] suggest the majority of the identified proteins likely originate from the ER membrane, rather than mitochondria/peroxisomes. Moreover, proteins with a single, N-terminal helix or a signal peptide were more common (10 out of 16 identified proteins with at least one transmembrane segment or signal peptide, 4 of which are type II membrane proteins) than targets with a single C-terminal helix (2 non-mitochondrial TA-proteins and 1 with a C-terminal helix). The remaining 3 targets are present in the inner mitochondrial membrane (with the helix placed in the N-terminus for 2 targets or in the middle for 1 that has a more elusive biogenesis. Taken together, this strengthens the hypothesis that P5A-ATPases transport a broad range of helices, including type II proteins as found here and previously as well as TA-proteins as identified earlier, and likely beyond as also suggested by our E2P and E2.P$_i$ structures. Multi-pass transmembrane membrane proteins as cargo are also possible[39], as such were identified in our MS analysis, and because of the ER luminal extension that reenters the membrane in our E2P structure. Interestingly, a broad cargo range of P5A-ATPase has already been proposed, where they have been suggested to assist in controlling that most eukaryotic membrane proteins are positively charged on the 'inside' (cytoplasm), similarly to their prokaryotic counterparts[40].

## Discussion

With the ER membrane helix removal/dislocase/flippase mechanism previously proposed for P5A-ATPases, uptake from the ER membrane is expected in the E1P/E2P transition, in agreement with the detected cavity in our E2P and E2.P$_i$ structures (Fig. 6)[23,37,38]. In the E2P stage, E458 and E462, in the center of the cleft, would be disruptive for binding of a very hydrophobic helix, but the remainder of the transport pathway is largely hydrophobic, which may imply binding of partially hydrophobic helices. Alternatively, an unexplained cryo-EM density associated with E458 in the E2.P$_i$ structure may relate to a feature that serves to stabilize the binding (Supplementary Fig. 15). The negatively charged P5A-residues on the ER lumen side could help stabilize any charged or polar residues in the tail at the membrane interface of the cargo, including mis-inserted membrane proteins that disobey the "positive-inside rule"[64]. The closing of the ER luminal end of the cleft would then push the helix into the cytoplasm, and following release, the cytoplasmic cavity is capped by the Plug-domain, as observed in our E1 structure. As phosphorylation is triggered, the Plug-domain is released from its blocking position, and uptake from the ER membrane can then be established in a late E1P, or E2P/E2.P$_i$ stage. The unexplained features observed in the E1P-ADP and E1P structures could then relate to cargo that just has been released by, or that is about to be taken up by the ATPase, respectively, although the exact sequence of events does not match the states assigned to the structures.

Then what about the insertion/translation/secretion mechanism previously proposed for P5A-ATPases such as ScSpf1, CATP-8, and hATP13A1[16,18,22,35,41,42]? While it is tempting to interpret the observations of these studies as coupled to the ER quality-control dislocation machinery that would indirectly enhance insertion/translation/secretion, a direct role for P5A-ATPases in such export from the cytoplasm cannot be excluded based on the available structural data. Indeed, the native (correct) membrane topology of type II generally agrees with the orientation of the helix detected in our E2.P$_i$ structure (with the N-terminus facing the cytosol). Furthermore, redeeming incorrectly inserted proteins (with inverse topology) is in certain cases associated with the challenge of re-importing to the cytosol a long stretch of soluble amino acids that may even have folded as a domain (such as for type III proteins that constitute a subset of the here identified putative clients). Rearrangement from the E1 state would then provide access to the cytoplasmic pocket, thereby permitting the initial binding of the cargo (Fig. 6). In this scenario, the feature observed in the E1P-ADP$^{membranous-feature}$ structure may represent a spontaneously inserted signal peptide, although such was not identified by our MS analysis (also not in membrane proteins with more than one helix). Cargo binding would stimulate conformational changes to E2P$^{cargo}$, accompanied by a complete opening of the cleft in the M-domain, permitting amino acids to protrude into the membrane environment. We note that the architecture of the cleft may accommodate hydrophobic polypeptides, particularly if they lack secondary structure as can be expected during insertion, as the polar backbone of the polypeptide may be stabilized by the amphipathic cleft of the P5A-ATPase, while hydrophobic portions can interact with residues of similar uncharged nature in the cleft or with the surrounding lipid bilayer. The establishment of a secondary structure (folding) would be accompanied by hydrophobic residues being expelled from the transport pathway, thereby decreasing the ATPase affinity for the cargo. Once the entire helix is formed in the cleft, it would be released to the lipid bilayer following the E2.P$_i$$^{cargo}$ configuration, expelled from the ATPase due to conflicting charges and instead preferring the hydrophobic lipid bilayer. Following the release, the Plug-domain would ensure that passage across the membrane is prevented. Future more detailed studies will be needed to further dissect how the P5A-ATPases operate at the molecular level. A major consequence of the available mechanisms, regardless of transport direction, is that they appear to be out of phase compared to other P-type ATPases, for example, with the cytosol-open E1P-ADP/E1P structures and the all-through cleft in the E2.P$_i$ state.

In summary, we have expanded the repertoire of structures available of P5A intermediates, revealing membrane penetrating E2P and E2.P$_i$ states, as well as blocked and accessible inward-open E1/E1-ATP and E1P-ADP/E1P configurations. This has provided further details on how the cargo interacts with the transporter in E2P/E2.P$_i$ and perhaps in the E1P-ADP/E1P states. Furthermore, our data point towards a cardinal role of the Plug-domain for transport, as it serves as a turnover-dependent gate of the M-domain and is directly linked to critical parts of the soluble A- and P-domains, thereby coupling the soluble and membrane-spanning portions. Nonetheless, several questions regarding P5A transport and its effects on (patho)physiology remain. The cargo and possible counter cargo specificity remain to be further resolved—what determines if and how cargo is targeted by P5A-ATPases; how hydrophobic is the segment that traverses the cleft, and is it followed by positively charged residues? Clearly, the open cleft in the E2P/E2.P$_i$ states, and the cryo-EM densities in the transport pathway that we interpret as cargo clearly speak for the involvement of a large cargo such as helices. Moreover, the bulk of the available data supports a dislocase role for P5A-ATPases, in quality control to resurrect ER membrane mis-inserted proteins and eventually increase the correctly inserted ratio, although the transport directionality cannot be conclusively assigned based on the available structural information. Consequently, critical gaps remain also regarding the transport mechanism. The exact role of the Plug-domain needs to be further investigated. It may also be responsible for a physical link between the P5A-ATPase and other proteins involved in translocation, as co-localization studies and genetic data suggest the Sec61/Sec62/Sec63 translocon and P5A-ATPase are coupled[65]. Studies of uptake, binding, and release of designated target proteins, and not averages of multiple different cargo from native sources, will likely also be necessary to fully decipher the molecular details of the transport mechanism.

## Methods

### Cloning

The full-length gene coding for CtSpf1 (Uniprot: G0S4Z4) with two introns was amplified from the genomic DNA of *Chaetomium thermophilum* (DSM1495) with primers CtSpf1-22b-F and CtSpf1-22b-R. The PCR products were cloned into the pET-22b vector using the

NEBbuilder HiFi DNA Assembly Master mix, yielding construct pET-22b-CtSpf1$^{introns}$. The two introns were individually removed with primer pairs Δintron1-F/Δintron1-R and Δintron2-F/Δintron2-R by using construct pET-22b-CtSpf1$^{introns}$ as the template, yielding the pET-22b-CtSpf1 construct. To facilitate protein expression and purification, the C-terminus of CtSpf1 (with the last two arginines removed) was tagged with a TEV cleavage site, a short linker containing residues GGGGS, and green fluorescence protein (GFP) with a 10xHis tag. This was cloned into the galactose-induced vector pEMBLyex4[66] with the CtSpf1-C-GFP-F/CtSpf1-C-GFP-R primer pair by using the pET-22b-CtSpf1 construct as a template, yielding the pEMBLyex4-CtSpf1-TEV-G4S-GFP-His10 construct. All constructs were confirmed by sequencing, and the primers are listed below.

CtSpf1-22b-F: TCTTTATTTTCAGGGCATGGCGCCGCTCGTTGATA ATCCGCA CtSpf1-22b-R: TTAGCAGCCGGATCTCATTACCGCCTCTG GGCCCATTGCTGCT Δintron1-F: CGCGACAAGGTTGGCGACAACAAGA CGAACATCTC Δintron1-R: TGTTGTCGCCAACCTTGTCGCGGACCAGC TTG Δintron2-F: CAGATCGAGCCTCGCACCGAAGTAATTGACCTTGA Δintron2-R: CTTCGGTGCGAGGCTCGATCTGGTCGCACAGCT CtSpf1-C-GFP-F: CAATTCTAAGATAATTATGGCGCCGCTCGTTGATAATC CtSpf1-C-GFP-R: ATTGAAAATACAAATTTTCCTGGGCCCATTGCT GCTGTTG CtSpf1(D499N)-F: GCCTGCTTTAACAAAACTGGAACTC CtSpf1(D499N)-R: AGTTTTGTTAAAGCAGGCGACGTC

## Protein production

The expression plasmid pEMBLyex4-CtSpf1-TEV-G4S-GFP-His10 was transformed into the PAP1500 *S. cerevisiae* strain[66] using the LiAc/SS carrier DNA/PEG method[67] and plated on SD media (20 g/L glucose, 1.9 g/L Yeast Nitrogen Base, 5 g/L (NH$_4$)$_2$SO$_4$, 60 mg/L leucine, 30 mg/L lysine) supplied with 15 g/L agar, using uracil selection and grown for 3 days at 30 °C. Single colonies were inoculated in 5 mL SD media at 30 °C for 24 h, shaking at 200 rpm. The cells were subsequently pelleted and transferred to 250 mL SD media without leucine and incubated for 24 h at 30 °C with shaking. Next, 50 mL preculture was cultivated in 800 mL expression media (3 % v/v glycerol, 5 g/L glucose, 1.9 g/L Yeast Nitrogen Base, 5 g/L (NH$_4$)$_2$SO$_4$, 1.17 g/L -Ile, -Ura dropout amino acid mixture) for 24 h at 30 °C with shaking. The temperature was reduced to 25 °C and continued culturing for 2 h, then 200 mL induction media (3 % (v/v) glycerol, 20 g/L galactose, 1.9 g/L Yeast Nitrogen Base, 5 g/L (NH$_4$)$_2$SO$_4$, 1.17 g/L -Ile, -Ura dropout amino acid mixture) was added to induce protein expression. The cells were harvested 24 h following induction and washed with lysis buffer (10 mM Tris-HCl pH = 7.5, 60 mM NaCl, 4 % (v/v) glycerol, 2 mM 2-β-mercaptoethanol (BME)). Usually, 3 g cell pellets were obtained from 1 L cell culture.

## Protein purification

To capture different biological states, individual batches of cells were cultured and prepared for different samples. The cells were disrupted by high-pressure homogenization (Xpress) and resuspended in lysis buffer at 50 mg/mL. Unbroken cells and cell debris were removed by centrifugation at 4000×*g* for 10 min, and crude membranes were isolated by 2 h centrifugation at 165,000 × *g*. Next, crude membranes were solubilized with solubilization buffer (50 mM Tris-HCl pH=7.5, 500 mM NaCl, 10 % (v/v) glycerol, 2 mM BME, 2% (w/v) n-dodecyl-β-maltoside and 0.2% (w/v) cholesteryl hemisuccinate (CHS)) to a final concentration of 50 mg/mL at 18 °C for 2 h. The supernatant was collected by centrifugation at 190,000 × *g* for 30 min. The protein was initially purified using a 5 mL Histrap immobilized metal affinity chromatography (IMAC) column (Cytiva) equilibrated with buffer A (20 mM Tris-HCl pH = 7.5, 150 mM NaCl, 10% (v/v) glycerol, 2 mM BME, 0.05% (w/v) DDM and 0.002% (w/v) CHS). The column was washed with 20 column volumes of buffer A supplied with 60 mM imidazole to remove contaminants. The target protein was eluted with buffer A with 300 mM imidazole. The elution fractions were concentrated and applied to a Superose6 size-exclusion chromatography (SEC) column (Cytiva) for further purification with buffer A. The protein purity was checked using SDS-PAGE. The peak fractions were pooled and concentrated to around 5 mg/mL for downstream processing. Different intermediate states were obtained based on this procedure, with modifications described below.

The E2P state was treated with 2 mM BeF$_3^-$ and 5 mM MgCl$_2$, supplemented during cell lysis, and then with 1 mM BeF$_3^-$ and 2 mM MgCl$_2$ in all subsequent buffers.

The E2.P$_i^{cargo}$/E1P$^{cytosolic-feature}$ states were treated with 5 mM AlF$_4^-$ and 10 mM MgCl$_2$, supplemented during cell lysis, and preserved in all subsequent buffers.

The E1P-ADP state was treated with 2 mM AlF$_4^-$ and 5 mM MgCl$_2$, supplemented during cell lysis and preserved in all subsequent buffers.

The E1 (apo) state was incubated at 50 °C for 10 min in a water bath prior to membrane isolation.

The E1-ATP state was treated with 1 mM AMP-PNP and 1 mM MgCl$_2$, supplemented with the nanodisc sample in the final step before grid preparation.

## Nanodisc reconstitution

MSP1D1 and MSP1E3D1 were purified as previously reported[9,68], with the His tag removed. Yeast polar lipids (Avanti Lipids) were prepared in 20 mM Tris-HCl pH = 7.5, 100 mM NaCl, 0.5% (w/v) DDM for the nanodisc reconstitution. For the E1-ATP state, the purified protein was reconstituted into MSP1D1 nanodisc with a molar ratio of CtSpf1-GFP:MSP1D1:lipids of 1:4:100. All other states were prepared with MSP1E3D1, reconstituted with a molar ratio of CtSpf1-GFP:MSP1E3-D1:lipids of 1:5:100. 200 mg bio-beads SM2 (Bio-Rad) were used to remove detergent for protein incorporation into the nanodiscs. Empty disks were removed through IMAC, and protein-containing nanodiscs were further purified by SEC using a Superdex 200 column (Cytiva) with running buffer 20 mM Tris-HCl pH = 7.5, 150 mM NaCl for the E1 (apo) and E1-ATP states. For the E2P state, the running buffer was further supplied with 1 mM BeF$_3^-$ and 2 mM MgCl$_2$. For the other states, the running buffer was instead further supplied with 2 mM AlF$_4^-$ and 2 mM MgCl$_2$. The relevant nanodisc peak fractions following SEC were pooled and concentrated to 5–10 mg/mL for cryo-EM grid preparation.

## Cryo-EM grid preparation

For the E1-ATP state, 1 mM AMP-PNP and 1 mM MgCl$_2$ were incubated with a protein sample on ice for 30 min before freezing the grids. For the E1P-ADP state, 1 mM ADP was incubated with 2 mM AlF$_4^-$ purified sample (as also supplemented from cell lysis) on ice for 30 min before freezing. To improve the particle distribution, 0.5 mM or 1 mM fluorinated fos-choline-8 was added to each sample prior to grid preparation. Quantifoil holey carbon grids (R1.2/1.3 copper; 300 mesh) were glow-discharged for 40 s. The grids were prepared using a Vitrobot Mark IV operated at 100% humidity and 4 °C. 3 μL of protein sample was applied to each grid, incubated for 5 s, blotted for 3 s, and then plunged frozen into liquid ethane. Frozen grids were stored in liquid nitrogen until data collection.

## Cryo-EM data acquisition

The E1-ATP dataset was collected on a Titan Krios electron microscope (FEI) operated at an acceleration voltage of 300 kV with a Falcon3 detector in counting mode and a pixel size 0.832 Å. The total dose was 40 e/Å$^2$ in 40 frames. The datasets for all the other states were collected on a Titan Krios electron microscope operated at an acceleration voltage of 300 kV, and images were recorded on a Gatan K3 detector with an applied energy filter of 20 eV. The E1 (apo) and E2P (BeF$_3^-$) datasets were collected using the super-resolution mode with pixel size 0.55 Å and a total dose of 50 e/Å$^2$ in 40 frames. The E1P-ADP dataset was collected using super-resolution mode bin2 with pixel size 0.846 Å and a total dose of 50 e/Å$^2$ in 40 frames. The E2.P$_i$ (AlF$_4^-$)

dataset was collected using the super-resolution mode bin2 with pixel size 0.8464 Å and a total dose of 50.2 e/Å$^2$ in 40 frames.

## Cryo-EM data processing

All data processing was performed with cryoSPARC[69], and the movies were initially processed using Path motion correction with 2× binning for the E1 (apo) and E2P datasets. Patch CTF was applied for the contrast transfer function estimation. Bad micrographs, such as those that were empty, broken, or containing ice, were removed through manual inspection. Selected suitable micrographs were subjected to further analysis.

For the E2P state, 869,782 particles were initially picked from 1479 of 4777 selected micrographs through blob picking. The particles were extracted and subjected to 2 rounds of 2D classification to generate templates. The selected templates were used for template picking, and a total of 2,307,520 articles were picked from the 4777 micrographs. Extracted particles were subjected to 3 rounds of 2D classification, resulting in 763,922 picks. 80,000 particles were used for the ab initio reconstruction, with 5 generated classes, followed by heterogeneous refinement with all particles. The second-round heterogeneous refinement was performed with selected particles from class 0 and class 3, which clearly show protein features. The final set of 249,499 particles and the resulting model was subjected to non-uniform refinement[70] and CTF refinement, yielding a map with an overall resolution of 3.5 Å, based on an FSC 0.143. The local resolution was estimated using Cryosparc with an FSC 0.143 cut-off. The processing flow chart is shown in Supplementary Fig. 3.

For the E2.P$_i$$^{cargo}$ and E1P state that 645,581 particles were initially picked from 1438 of 6451 micrographs through blob picking. The particles were extracted and subjected to 2 rounds of 2D classification to generate templates. The selected templates were used for template picking, and a total of 2,849,415 particles were picked from 6451 micrographs. Extracted particles were subjected to 4 rounds of 2D classification, resulting in 495,256 picks. In total, 50,000 particles were used for the ab initio reconstruction, with 5 classes generated, followed by heterogeneous refinement with all particles. A second-round heterogeneous refinement was performed with selected particles from class 0 and class 4, respectively. The final set of 185,340 particles from class 0 and the resulting map was subjected to non-uniform refinement and CTF refinement, yielding a map with an overall resolution of 3.4 Å based on an FSC 0.143. The final set of 107,529 from class 4 and the resulting map was subjected to non-uniform refinement and CTF refinement, yielding an E1P map with an overall resolution of 3.7 Å based on an FSC 0.143. The local resolutions for two states were estimated using Cryosparc with an FSC 0.143 cut-off. The processing flow chart is shown in Supplementary Fig. 7.

For the E1 (apo) state, 832,472 particles were initially picked from 1675 of 4081 micrographs through blob picking. The particles were extracted and subjected to 2 rounds of 2D classification to generate templates. The selected templates were used for the template picking and a total of 1,621,181 particles were picked from the 4081 micrographs. Extracted particles were subjected to 3 rounds of 2D classification, resulting in 648,009 picks. 70,000 particles were used for ab initio reconstruction, with 5 classes generated, followed by heterogeneous refinement with all particles. The second-round heterogeneous refinement was performed with selected particles from class 2 and class 3. The final set of 238,138 particles and the resulting map was subjected to non-uniform refinement and CTF refinement, yielding map1 with an overall resolution of 3.4 Å based on an FSC 0.143 and 191,389 particles. The resulting map was subjected to non-uniform refinement and CTF refinement, yielding map2 with an overall resolution of 3.5 Å based on an FSC 0.143. The local resolutions for the two maps were estimated using Cryosparc with FSC 0.143 cut-off. The processing flow chart is shown in Supplementary Fig. 8.

For the E1-ATP state, 1,042,973 particles were picked from 3231 micrographs through blob picking. Local motion correction was performed, and particles were extracted. Particles were subjected to 3 rounds of 2D classification, with 182,187 particles obtained. 30,000 particles were used for the ab initio reconstruction, with 3 classes generated, followed by heterogeneous refinement with all particles. The final set of 115,476 particles and the resulting map was subjected to non-uniform refinement, yielding a map with an overall resolution of 3.2 Å based on an FSC of 0.143. The local resolution was estimated using Cryosparc with an FSC 0.143 cut-off. The processing flow chart is shown in Supplementary Fig. 11.

For the E1P-ADP state, 395,731 particles were initially picked from 2533 of 7096 micrographs through blob picking. The particles were extracted and subjected to 2 rounds of 2D classification to generate templates. The selected templates were used for the template picking and a total of 2,856,624 particles were picked from 7096 micrographs. Extracted particles were subjected to 3 rounds of 2D classification, resulting in 449,894 picks. Fifty-thousand particles were used for the ab initio reconstruction, with 5 classes generated, followed by heterogeneous refinement with all particles. A second-round heterogeneous refinement was performed with selected particles and four maps since two of them (class 0 and class 1) were almost identical. The final set of 151,338 particles and the resulting map was subjected to non-uniform refinement and CTF refinement, yielding the E1P-ADP$^{membranous-feature}$ map with an overall resolution of 3.4 Å, based on an FSC 0.143. In parallel, a second final set of 171,420 particles and the resulting map of Class 3 were subjected to non-uniform refinement and CTF refinement, yielding the E1P-ADP map with an overall resolution of 3.4 Å, based on an FSC 0.143. The local resolutions for the two maps were estimated using Cryosparc with FSC 0.143 cut-off. The processing flow chart is shown in Supplementary Fig. 12.

## Model building and refinement

A CtSpf1 model was generated using the SWISS-MODEL online server[71], using the sequence of CtSpf1 and the cryo-EM structure of ScSpf1 (PDB-ID 6XMP) as templates. The generated CtSpf1 model was initially fitted into the best quality cryo-EM density map of E1-ATP using USCF chimera. De-novo model building, guided by the density of bulky side chains, was conducted in Coot[72]. Cycles of model-building adjustment in Coot and real-space refinement using real_space_refine[73] in Phenix using the sharpened E1P-ATP map were performed to obtain the final model. The E1-ATP model starts with residue Ala2 in the N-terminus. Amino acids 118–124 of the NTD-domain, 357–365 of the A-domain, 522–525 and 593–597 of the N-domain, 882–1011 of the Plug-domain, 1145–1150 of the loop connecting M7 and M8, and 1286–1326 of the C-terminus were left as unstructured loops due to poor density. The model building of the E1 (apo), E1P (AlF$_4^-$), and E1P-ADP (AlF$_4^-$–ADP) states were facilitated using our E1P-ATP model, each fitted into the corresponding sharpened maps. Next, the models were manually adjusted in Coot and refined using real_space_refine in Phenix. The model building of the E2P (BeF$_3^-$) and E2.P$_i$ (AlF$_4^-$) conformations were carried out with the generated Swiss model and the structure of ScSpf1 (PDB-ID 6XMT) as templates, which were fitted into the corresponding sharpened map as a rigid body. The strategies of the model adjustment and refinement were similar ass for the above-mentioned E1-ATP state. The clear feature of endogenous cargo density in E2.P$_i$$^{cargo}$ (AlF$_4^-$) was modeled with 31 residues (as a poly-alanine), with the N-terminus in the cytoplasm. Model validation was performed using MolProbity[74]. The figures were prepared with UCSF ChimeraX[75], UCSF Chimera[76], and Pymol.

## In solution-digestion

The protein samples for mass spectrometry were isolated as three independent biological replicates (three separate expressions and purifications) using the above-mentioned procedures with the

following small modifications. The BeF$_3^-$ (1 mM) was added from the cell lysis and maintained during the entire purification. For the IMAC, the protein samples were purified using Ni-NTA agarose resin. Twenty microgram proteins from the IMAC elution were prepared for liquid chromatography-mass spectrometry (LC-MS) by reduction in 10 μM dithiothreitol (DTT), alkylation by 20 μM iodoacetamide (IAA) followed by trypsination at trypsin:protein ratio 1:50 o/n at 37 °C. Peptide desalting was performed using Ultra Microspin column Silica C18 (SUM SS18V, 3–30 mg capacity, The Nest Group Inc., South Borough) following the instructions recommended by the manufacturer. Peptides were dried by SpeedVac and resolved in 2% acetonitrile (ACN) and 0.1% trifluoroacetic acid (TFA) to 0.5 μg/μl.

## Mass spectrometry acquisition

The LC–MS detection was performed on an Orbitrap Fusion Tribrid mass spectrometer equipped with a Nanospray Flex ion source and coupled with an EASY-nLC 1000 ultrahigh pressure liquid chromatography (UHPLC) pump (Thermo Fischer Scientific). One microgram of protein digest was injected into the LC–MS. Peptides were concentrated on an Acclaim PepMap 100 C18 precolumn (75 μm × 2 cm, Thermo Scientific, Waltham, MA) and then separated on an Acclaim PepMap RSLC column (75 μm × 25 cm, nanoViper, C18, 2 μm, 100 Å) at 40 °C and with a flow rate of 300 nL/min. Solvent A (0.1% formic acid in water) and solvent B (0.1% formic acid in acetonitrile) were used to create a nonlinear gradient to elute the peptides. For the gradient, the percentage of solvent B was maintained at 3% for 3 min, increased from 3 to 25% for 60 min, then increased to 60% for 10 min, and then increased to 90% for 2 min and then kept at 90% for another 8 min to wash the column.

The Orbitrap Fusion was operated in the positive data-dependent acquisition (DDA) mode. The peptides were introduced into the MS via stainless steel Nano-bore emitter (OD 150 μm, ID 30 μm) with a spray voltage of 2 kV, and the capillary temperature was set at 275 °C. Full MS survey scans from m/z 350–1350 with a resolution of 120,000 was performed in the Orbitrap detector. The automatic gain control (AGC) target was set to $4 \times 10^5$ with an injection time of 50 ms. The most intense ions (up to 20) with charge states 2–5 from the full scan MS were selected for fragmentation in the Orbitrap. The precursors were isolated with a quadrupole mass filter set to a width of 1.2 m/z. Precursors were fragmented by high-energy collision dissociation (HCD) at a normalized collision energy (NCE) of 30%. The resolution was fixed at 30,000, and for the MS/MS scans, the values for the AGC target and injection time were $5 \times 10^4$ and 54 ms, respectively. The duration of dynamic exclusion was set to 45 s, and the mass tolerance window was 10 ppm.

## Mass spectrometry data analysis

The raw data files were subjected to database searches against G0S4Z4 from *Chaetomium thermophilum* and the entire *Saccharomyces cerevisiae* Uniprot proteome (strain ATCC 204508/S288c) downloaded on 20220615, using Maxquant 2.0.2[77], with the following parameters: precursor tolerance: 20 ppm (first search) 4.5 ppm (main search); fragment tolerance: 20 ppm; enzyme: trypsin (specific) with maximum 2 missed cleavages; variable modifications: methionine oxidation and N-terminal acetylation; fixed modifications: carbamidomethylation of cysteine. The protein and peptide-level false discovery rate was set to 0.01. Post-processing was performed in Perseus 1.6.15. Common contaminants, including keratins, were filtered out, and only proteins that were identified in 3 out of 3 replicates and contained at least 2 matched peptides were taken into consideration for comparisons between datasets.

## The functional complementation assay

A *ScSpf1*-deleted and WT BY4741 (*MATa: ura3Δ0 leu2Δ0 his3Δ1 met15Δ0*) yeast strain was used for the complementation assay. The full-length *ScSpf1* gene was cloned from the genome of the BY4741 strain and introduced into the pEMBLyex4 vector with a C-terminal GFP tag. The GFP only construct (also in pEMBLyex4) was transformed into the wild-type (WT) and *Spf1*-deleted BY4741 strains, and ScSpf1-GFP, ScSpf1(D487N)-GFP (D487N in ScSpf1 is equivalent to D499N in CtSpf1), CtSpf1-GFP and CtSpf1(D499N)-GFP were separately transformed into the Spf1-deletion BY4741 strain using the Li-Ac method. Positive transformants were selected on SC plates without uracil (20 g/L glucose, 1.9 g/L Yeast Nitrogen Base, 5 g/L (NH$_4$)$_2$SO$_4$, 60 mg/L leucine, 30 mg/L lysine, 1.17 g/L -Ile, -Ura dropout amino acid mixture, 20 g/L agar). Single colonies were inoculated in SC medium for 16 hours at 30 °C and were then inoculated in fresh SC medium (without additional leucine and lysine) to reach to log phase with OD around 0.8. The cells were serially 10-fold diluted with YP medium (1% w/v yeast extract and 2% w/v bacteriological peptone). 6 μL cell drops were spotted onto the YPdG plates (1% w/v yeast extract, 2% w/v bacteriological peptone, 0.5% w/v D-glucose, 2% w/v galactose, and 2% w/v agar) containing defined amounts of caffeine (0 or 7.5 mM). The plates were incubated at 30 °C for 2 days for imaging.

## Reporting summary

Further information on research design is available in the Nature Portfolio Reporting Summary linked to this article.

## Data availability

The data that support this study are available from the corresponding authors upon request. Cryo-EM maps have been deposited in the Electron Microscopy Data Bank (EMDB) under accession codes EMD-17039 (E1), EMD-17040 (E1-ATP), EMD-17041 (E1P-ADP^membranous-feature^), EMD-17042 (E1P^cytosolic-feature^), EMD-17043 (E2P^cargo^) and EMD-17044 (E2.P$_i$^cargo^). The atomic coordinates have been deposited in the Protein Data Bank (PDB) under accession codes 8OP3 (E1), 8OP4 (E1-ATP), 8OP5 (E1P-ADP^membranous-feature^), 8OP6 (E1P^cytosolic-feature^), 8OP7 (E2P^cargo^) and 8OP8 (E2.P$_i$^cargo^). The mass spectrometry proteomics data have been deposited to the ProteomeXchange Consortium via the PRIDE partner repository with the dataset identifier PXD042401. The previously published PDB codes referred to in the paper under accession codes 6XMT, 4H1W, 3N8G, 6K7G, 6XMP, 7N75, and 6XMU.

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

## Acknowledgements

This study was supported by the Swedish Research Council (2016-04474 and 2022-01315; P.G.), Knut and Alice Wallenberg Foundation (2015.0131 and 2020.0194; P.G.), The Lundbeck Foundation (R322-2019-2588; VB, R324-2019-1855; KW, R218-2016-1548, R133-A12689, R313-2019-774 and R346-2020-2019; PG), the Danish Council for Independent Research (9039-00273 and 3101-00303; P.G.), Carl Trygger Foundation (CTS 21:1773), the Crafoord Foundation (20170818, 20180652, 20200739 and 20220905; PG) and The Per-Eric and Ulla Schyberg Foundation (38267; P.G.). P.L. also obtained support from the Royal Physiographic Society of Lund. The funders had no role in study design, data collection and analysis, decision to publish, or preparation of the paper. We would like to thank Julian Conrad, Karin Wallden, Dustin Morado, and Marta Corrani at the Cryo-EM Swedish National Facility in Stockholm for sample screening and data collection. The Cryo-EM Swedish National Facility at SciLifeLab is funded by the Knut and Alice Wallenberg, Family Erling Persson and Kempe Foundations, SciLifeLab, Stockholm University, and Umeå University. We also would like to thank Tillmann Hanns Pape at the Danish Cryo-EM Facility at the Core Facility for Integrated Microscopy (CFIM) at the University of Copenhagen for sample screening and data collection. The Danish Cryo-EM Facility at CFIM, University of Copenhagen, is supported by Novo-Nordisk Foundation grant id NNF14CC0001. We also acknowledge the Lund University Cryo-EM platform for access to the GPU computers. We would like to thank Profs. Per Amstrup Pedersen (University of Copenhagen) and Kefeng Lu (Sichuan University) for providing the PAP1500 and *ScSpf1* knockout strains, respectively, and Kefeng Lu also for the cellular localization images. We also would like to thank Charlotte Welinder and Hong Yan at the Center for Translational Proteomics at the Medical Faculty, Lund University, for their assistance with the M.S.

## Author contributions

P.L., K.W., and P.G. conceived the study. P.L. and P.G. supervised the project. P.L. designed the experiments, performed the cloning, established and optimized protein overproduction and purification, and designed the nanodiscs-reconstitution for cryo-EM structural studies. KW prepared the grids for generation of the E1-ATP, E1P-ADP, E2.P$_i$$^{cargo}$/E1P$^{cytosolic-feature}$ states, K.W. collected the E1-ATP state dataset at CFIM of the University of Copenhagen. All other state datasets were collected at SciLifeLab in Stockholm, assisted by P.L. P.L., who processed the cryo-EM datasets and built and refined the structures. P.L. designed the MS experiments and prepared the samples for the MS. V.B., and P.M.H. conducted the MS analysis. P.L. conducted the complementation assay. P.L. prepared the figures, except for those related to the MS analysis (V.B. and P.M.H.) and the distance difference matrix (V.B.). P.L. conducted the initial data analysis and wrote the first draft. P.L., V.B., K.L.-P., and P.G. further conducted data analysis and interpretation. P.L., V.B., and P.G. finalized the paper. All authors commented on the paper.

## Funding

## Competing interests

The authors declare no competing interests.
