## [Peer Review file · Nature Communications]

The structure and function of P5A-ATPases

Corresponding Author: Dr Pontus Gourdon

This manuscript has been previously reviewed at another journal. This document only contains reviewer comments, rebuttal and decision letters for versions considered at Nature Communications.

Version 0:

Reviewer comments:

Reviewer #1

(Remarks to the Author)

The manuscript by Pin Li et al. reports cryo-EM structures of the CtSpf1 P5A-ATPase in various transport cycle states. These states were stabilized using ligands previously employed for other P-ATPases. The overall strategy adopted here mirrors that of McKenna et al. (2020) for the ScSpf1 in that they used the averaged data from a mixture of cargo molecules in their native source. Possibly for this reason, neither this study nor the previous one definitively unambiguously identifies non-ATPase densities that are critical components of the proposed P5A-ATPase function. Despite this limitation, the manuscript advances our understanding of P5A-ATPases by providing structures of the protein in novel intermediate states and describing with more detail features like the plug structure, which were absent in the preceding study of the ScSpf1. Furthermore, the identification of new densities in the E2P and E1P-ADP structures of CtSpf1 is intriguing and leads the authors to propose a potential countertransport or insertion function alongside the dislocase activity. Unfortunately, the inability to identify these densities renders the model quite speculative.

Points for consideration:

- 1) CtSpf1-GFP Functionality (30°C): The manuscript utilizes the P5A-ATPase CtSpf1 from the thermophilic fungus *Chaetomium thermophilum*, terminally fused with green fluorescent protein (GFP) and overexpressed in *S. cerevisiae*. To strengthen the study, please provide experimental evidence demonstrating CtSpf1-GFP functionality in *S. cerevisiae* specifically at 30°C.
- 2) A previous study of ScSpf1 in detergent micelles shows that it has an uncoupled ATPase activity and that this activity correlates with the reaction temperature. Could some of the structural findings of the present study be related to the different working temperatures of CtSpf1 compared with ScSpf1? For example the poor conservation of the plug or the C-terminal helix that the authors find contacting the N-domain and seem exclusive of CtSpf1. Please consider addressing this issue in the discussion.
- 3) Mass spec data. The list of identified proteins in Supplemental Table 3 is currently grouped by the number of transmembrane helices (1 vs. multiple). However, the order within each group appears arbitrary and seems to follow UniProt identifiers. Please add to the table the number of peptide hits and unique peptides and sort each group in descending order of hits.
- 4) Cargo interaction (lines 160-165)
 - a) The affirmation that the author's findings are inconsistent with previous suggestions regarding the importance of ER lumen flanking C-terminal sequence for P5A interaction with so-called tail-anchored (TA) membrane proteins destined for mitochondria seems overstated. Indeed mass spec did not find any mitochondrial TA protein in the author's samples.
 - b) It is not clear if the inconsistency the authors mention (line 163) refers specifically to TA membrane proteins or all putative substrates with hydrophobic segments. There would be also an inconsistency with the author's finding of a well-integrated cargo in E2P despite E458-E462. In connection with this, the proposed transport mechanism (lines 379-380) relies on an undefined density in E2.Pi. The current limitations in defining the density hinder a clear understanding of the specific function of E458 and E462.

5) Cargo Specificity (Lines 183-186 & 339-347): For more cautious interpretation, limit conclusions to the specific characteristics of the cargo observed. The unexpected orientation observed in the model does not necessarily contradict previous findings but adds complexity.

6) Unidentified Features (Line 252-262): The identities of E1P.ADPmembranous-feature and E1Pcytosolic-feature remain unknown. The authors acknowledge this limitation and refrain from modeling these features due to their poorly defined secondary structure. However, the abstract describes the feature in E1P-ADP as a helix while in contrast the possibility of the feature being lipid molecules isn't explicitly reflected in the current transport model. If possible, consider acquiring additional data to definitively identify these features.

Additional points

1) Lines 96-97. I couldn't find in Methods when the GFP was it removed. Please add if necessary.

Version 1:

Reviewer comments:

Reviewer #1

(Remarks to the Author)

In this revised version of the manuscript by Li et al., the authors have addressed some of the critical points raised by this reviewer, incorporated experimental data on the functional state of CtSpf1 overexpressed in Sc, and modified the text to make it more comprehensive, particularly regarding the differences between the structures of Sc and Ct Spf1. The authors explicitly state that they have made unsuccessful attempts to improve the resolution of their structures. Thus, despite the limited resolution of the critical regions in the structures this study presents novel findings and hypotheses, making a significant contribution to our understanding of P5A-ATPase transporters.

We thank the reviewer for the evaluation, suggestions and comments to improve our manuscript “The structure and function of P5A-ATPases”. We have now addressed all comments and amended the manuscript as outlined below, with the changes highlighted in yellow in the main manuscript. Remarks and questions from the reviewer are shown in black. Our responses are shown in blue.

Reviewer #1 (Remarks to the Author):

The manuscript by Pin Li et al. reports cryo-EM structures of the CtSpf1 P5A-ATPase in various transport cycle states. These states were stabilized using ligands previously employed for other P-ATPases. The overall strategy adopted here mirrors that of McKenna et al. (2020) for the ScSpf1 in that they used the averaged data from a mixture of cargo molecules in their native source. Possibly for this reason, neither this study nor the previous one definitively unambiguously identifies non-ATPase densities that are critical components of the proposed P5A-ATPase function. Despite this limitation, the manuscript advances our understanding of P5A-ATPases by providing structures of the protein in novel intermediate states and describing with more detail features like the plug structure, which were absent in the preceding study of the ScSpf1. Furthermore, the identification of new densities in the E2P and E1P-ADP structures of CtSpf1 is intriguing and leads the authors to propose a potential countertransport or insertion function alongside the dislocase activity. Unfortunately, the inability to identify these densities renders the model quite speculative.

We thank the reviewer for the evaluation of our work and for the positive overall assessment of our findings.

Points for consideration:

1) CtSpf1-GFP Functionality (30°C): The manuscript utilizes the P5A-ATPase CtSpf1 from the thermophilic fungus *Chaetomium thermophilum*, terminally fused with green fluorescent protein (GFP) and overexpressed in *S. cerevisiae*. To strengthen the study, please provide experimental evidence demonstrating CtSpf1-GFP functionality in *S. cerevisiae* specifically at 30°C.

We have now provided additional *in vivo* data to demonstrate CtSpf1 can functionally complement ScSpf1 in *S. cerevisiae* at 30 °C. See Supplementary Fig. 2 in the revised version.

2) A previous study of ScSpf1 in detergent micelles shows that it has an uncoupled ATPase activity and that this activity correlates with the reaction temperature. Could some of the structural findings of the present study be related to the different working temperatures of CtSpf1 compared with ScSpf1? For example the poor conservation of the plug or the C-terminal helix that the authors find contacting the N-domain and seem exclusive of CtSpf1. Please consider addressing this issue in the discussion.

We thank the reviewer for pinpointing this. A previous study on ScSpf1 has indeed indicated that the ATPase activity correlates with the reaction temperature, which may well be the case also for CtSpf1. We have revealed lipids bound to different parts of the TM domain in all states, which could explain why the NTD domain is important for the ATPase activity and regulated by lipids, and the temperature could of course affect the fluidity of such lipids. We also agree with reviewer that poor conservation of the Plug-domain could be a second factor explaining why differences are observed for the Plug-domain of CtSpf1 but not for ScSpf1. We have included a sentence on this topic in the section on the Plug-domain movements.

3) Mass spec data. The list of identified proteins in Supplemental Table 3 is currently grouped by the number of transmembrane helices (1 vs. multiple). However, the order within each group appears arbitrary and seems to follow UniProt identifiers. Please add to the table the number of peptide hits and unique peptides and sort each group in descending order of hits.

We have now reformatted the table according to the suggestions by the reviewer.

4) Cargo interaction (lines 160-165)

a) The affirmation that the author's findings are inconsistent with previous suggestions regarding the importance of ER lumen flanking C-terminal sequence for P5A interaction with so-called tail-anchored (TA) membrane proteins destined for mitochondria seems overstated. Indeed mass spec did not find any mitochondrial TA protein in the author's samples.

It is indeed interesting that we could not identify any mitochondrial TA protein in our mass spectrometry samples, even though we exploited the phosphate analog employed to capture the E2P structure with bound cargo for those experiments. The cargo density we observed in E2P state is well integrated in the cleft, and the associated loop in the ER lumen reenters in the hydrophobic environment. This would be unexpected for mitochondrial TA proteins which have positively charged C-termini, but it does not exclude that mitochondrial TA proteins also are clients, also because the cryo-EM density likely represents an average of different cargos. Accordingly, we have revised the manuscript according to the suggestions of the reviewer.

b) It is not clear if the inconsistency the authors mention (line 163) refers specifically to TA membrane proteins or all putative substrates with hydrophobic segments. There would be also an inconsistency with the author's finding of a well-integrated cargo in E2P despite E458-E462. In connection with this, the proposed transport mechanism (lines 379-380) relies on an undefined density in E2.P_i. The current limitations in defining the density hinder a clear understanding of the specific function of E458 and E462.

The inconsistency we mentioned refers to mitochondrial TA membrane proteins with hydrophobic segments and C-terminal positively charged tails, or proteins with a similar build-up, such as the equivalent of the human ABCG2. Conversely, type II membrane proteins that also have been proposed as clients for P5A-ATPases better agree with the observed extra cryo-EM density in the E2P structure.

Regarding the undefined density in the E2.P_i structure next to residues E458 and E462 of the P5A-ATPase specific PPELPME-motif. In CtSpf1, residues E458 and E462 are involved in the interaction with a well-integrated cargo helix segment in both the E2P and E2.P_i states. This may imply that the positively charged tail following a hydrophobic helix (as in mitochondrial TA proteins) is not interacting with the glutamates. Conversely, it may suggest that the feature that spans the membrane in the E2P and E2.P_i structures is not completely hydrophobic. Alternatively, the unexplained density in the E2.P_i structure next to residues E458 and E462 may serve to charge-stabilize the glutamates such that a more hydrophobic segment is able to bind to the cleft. We agree with the reviewer that the fact that this feature is left unassigned hinders a clear understanding of the specific function of E458 and E462, and hence, more detailed studies need to be carried out to understand the specific function of these residues.

5) Cargo Specificity (Lines 183-186 & 339-347): For more cautious interpretation, limit conclusions to the specific characteristics of the cargo observed. The unexpected orientation observed in the model does not necessarily contradict previous findings but adds complexity.

We have now revised these sections according to the suggestions of the reviewer. We agree that we cannot exclude cargo types, and that our data rather points to a broader client range.

6) Unidentified Features (Line 252-262): The identities of E1P.ADPmembranous-feature and E1Pcytosolic-feature remain unknown. The authors acknowledge this limitation and refrain from modeling these features due to their poorly defined secondary structure. However, the abstract describes the feature in E1P-ADP as a helix while in contrast the possibility of the feature being lipid molecules isn't explicitly reflected in the current transport model. If possible, consider acquiring additional data to definitively identify these features.

We agree with the reviewer and have changed the wording of the abstract. Moreover, we have attempted to acquire additional data to definitively identify these unassigned cryo-EM features, but these efforts have been fruitless.

Additional points

1) Lines 96-97. I couldn't find in Methods when the GFP was it removed. Please add if necessary.

The CtSpf1 samples used for cryo-EM included the GFP-tag (non-cleaved samples). We have now clarified this matter in the manuscript and methods sections.